# [^99m^Tc]Tc-Labeled Plectin-Targeting Peptide as a Novel SPECT Probe for Tumor Imaging

**DOI:** 10.3390/pharmaceutics14050996

**Published:** 2022-05-06

**Authors:** Jiali Gong, Lingzhou Zhao, Jiqin Yang, Meilin Zhu, Jinhua Zhao

**Affiliations:** 1Department of Nuclear Medicine, Shanghai General Hospital, Shanghai Jiao Tong University School of Medicine, Shanghai 200080, China; 18363992624@163.com (J.G.); zlz-330163.com@sjtu.edu.cn (L.Z.); 2Department of Nuclear Medicine, General Hospital of Ningxia Medical University, Yinchuan 750004, China; 3School of Basic Medical Sciences, Ningxia Medical University, Yinchuan 750004, China

**Keywords:** plectin, plectin-targeting peptide, [^99m^Tc]Tc, tumor imaging

## Abstract

Certain receptors are often overexpressed during tumor occurrence and development and closely correlate with carcinogenesis. Owing to its overexpression on the cell membrane and cytoplasm of various tumors, plectin, which is involved in tumor proliferation, migration, and invasion, has been viewed as a promising target for cancer imaging. Hence, plectin-targeting agents have great potential as imaging probes for tumor diagnosis. In this study, we developed a [^99m^Tc]Tc-labeled plectin-targeted peptide (PTP) as a novel single-photon emission computed tomography (SPECT) probe for tumor imaging and investigated its pharmacokinetics, biodistribution, and targeting ability in several types of tumor-bearing mouse models. The PTP had good biocompatibility and targeting ability to tumor cells in vitro and could be readily labeled with [^99m^Tc]Tc after modification with the bifunctional chelator 6-hydrazino nicotinamide (HYNIC). Furthermore, the prepared [^99m^Tc]Tc-labeled PTP ([^99m^Tc]Tc-HYNIC-PTP) showed high radiochemical purity and excellent stability in vitro. In addition, favorable biodistribution, fast blood clearance, and clear accumulation of [^99m^Tc]Tc-HYNIC-PTP in several types of tumors were observed, with a good correlation between tumor uptake and plectin expression levels. These results indicate the potential of [^99m^Tc]Tc-HYNIC-PTP as a novel SPECT probe for tumor imaging.

## 1. Introduction

Molecular imaging can reflect biological events at the cellular and molecular levels during the occurrence and progression of diseases and has been widely applied for cancer diagnosis [1,2,3]. Because of the overexpression or activation of specific receptors in the tumorigenesis process, various tumor-targeting ligands, including antibodies, affibodies, nanobodies, peptides, and small-molecule compounds, have been labeled with appropriate radionuclides in the past several decades to develop molecular imaging probes for positron emission tomography (PET) and single-photon emission computed tomography (SPECT) imaging [4,5,6,7,8,9]. Among them, radiolabeled peptides are the best candidates for clinical translation. Some have been successfully applied in clinical practice, such as arginine-glycine-aspartate (RGD) for integrin receptors in solid tumors and octreotide for somatostatin receptors in neuroendocrine tumors [10,11,12]. This type of receptor-targeted nuclear medicine imaging shows an enormous value for tumor diagnosis and management, including but not limited to detection, staging, noninvasive quantification of receptor expression, therapy response monitoring, risk stratification, and patient selection [12,13,14]. Therefore, developing tumor-targeting imaging probes based on specific receptors is important for clinical applications.

Plectin, a large scaffolding protein with a molecular weight of 500 kDa, is commonly expressed in mammalian tissues and cells and plays an important multifunctional role in signal transduction and maintaining tissue integrity [15]. In addition to normal physiological functions in the cytoplasm, plectin is mislocalized on the cell membranes of tumors and is involved in a series of cellular activities, including tumor proliferation, migration, and invasion [16,17]. Plectin overexpression was initially found in pancreatic ductal cell cancer (PDAC) and 60% of early PDAC precursor lesions (PanIN III lesions), with very low expression levels in normal pancreatic tissues and pancreatitis [18,19]. However, as lesions progress to invasive tumors, plectin expression synchronously increases, indicating that plectin is a specific biomarker for primary and metastatic human PDAC [20,21]. Subsequently, plectin was shown to be overexpressed in various tumors, including ovarian, lung, head and neck, esophagus, prostate, and colon tumors, and has been viewed as a promising target for cancer diagnosis and treatment [16,17,22,23,24,25,26].

To date, plectin-targeting antibodies and peptides have been used to construct tumor-targeting imaging agents and drug delivery systems [27,28,29,30,31,32,33]. A plectin-targeting peptide (KTLLPTP, PTP) selected by the phage-display peptide library has been widely used as a targeting ligand because of its strong binding affinity, high specificity, and ease of modification [19]. Based on these properties, PTP has been conjugated with several imaging moieties for cancer detection, including magnetic resonance and fluorescence imaging [21,30]. However, there is still a lack of research on radionuclide-labeled PTP as nuclear imaging agents, and their capability for tumor diagnosis remains to be assessed. Therefore, in this study, a [^99m^Tc]Tc-labeled PTP was designed and prepared as a novel SPECT probe for tumor imaging. Its imaging performance in pancreatic cancer and several types of tumors with different plectin expression levels was investigated. For [^99m^Tc]Tc radiolabeling, the N-terminus of PTP was modified with the bifunctional chelator hydrazinonicotinic acid (HYNIC) to synthesize [^99m^Tc]Tc-HYNIC-PTP. The synthesized probe had a high radiochemical purity (RCP) and excellent stability in vitro. More importantly, [^99m^Tc]Tc-HYNIC-PTP showed favorable biodistribution, fast blood clearance, and clear tumor accumulation in these different tumor models, and tumor uptake showed a good correlation with their plectin expression levels. Our data suggest the considerable potential of [^99m^Tc]Tc-HYNIC-PTP as a novel SPECT probe for tumor imaging.

## 2. Materials and Methods

### 2.1. Materials

The PTP was manufactured by ChinaPeptides Co., Ltd. (Shanghai, China). During the synthesis process, the N-terminus of PTP was modified with fluorescein isothiocyanate (FITC) or HYNIC to obtain FITC-PTP and HYNIC-PTP, respectively. Fetal bovine serum (FBS), RPMI 1640 medium, Cell Counting kit-8 (CCK-8), RIPA buffer, phenylmethanesulfonyl fluoride (PMSF), and bicinchoninic acid (BCA) protein assay kits were purchased from Shanghai Dobio Co., Ltd. (Shanghai, China). The [^99m^Tc]Tc-pertechnetate solution (Na[^99m^Tc]TcO_4_) was obtained from Shanghai Xinke Pharmaceutical Co., Ltd. (Shanghai, China). Tricine, ethylenediamine-N,N’-diacetic acid (EDDA), stannous chloride (SnCl_2_), acetonitrile (CH_3_CN), trifluoroacetic acid (TFA), phosphate-buffered saline (PBS), and Tris-buffered saline with Tween 20 (TBST) were purchased from Shanghai Macklin Biochemical Co., Ltd. (Shanghai, China). Other chemicals and solvents were supplied by Sinopharm Chemical Reagent Co. Ltd. (Shanghai, China).

### 2.2. Cells and Animals

PANC-1, BxPC-3, C6, U87, 4T1, A549, mouse β-TC-6 insulinoma cells, and human lung epithelial BEAS-2B cells obtained from the Chinese Academy of Sciences (Shanghai, China) were incubated in media and treated under the conditions recommended by the supplier. The animal experiments conformed to the National Institutes of Health Guidelines and were approved by the ethical committee of Shanghai General Hospital. Four-week-old female BALB/c nude mice (18–20 g) and healthy ICR mice (20–22 g) were purchased from the Shanghai Laboratory Animal Center of the Chinese Academy of Sciences (Shanghai, China). Animal models were established according to previously published protocols [34,35]. The mice were subcutaneously injected in their right-side flanks with 2 × 10^6^ C6, 1 × 10^7^ U87, 5 × 10^6^ 4T1, 1 × 10^7^ BxPC-3, or 1 × 10^7^ A549 cells suspended in 100 µL PBS. When the tumor nodules reached a 0.8–1.2 cm diameter, tumor-bearing mice were used for animal experiments.

### 2.3. Cytotoxicity Assay In Vitro and Toxicity Study In Vivo

CCK-8 assays were used to assess the potential cytotoxic effects of PTP on normal and tumor cell lines. Briefly, taking BxPC-3 cells as an example, the cells were seeded into 96-well plates at a density of 1 × 10^4^ per well with RPMI-1640 medium and 10% FBS. After 24 h, the medium was replaced with 100 µL of fresh medium containing different concentrations of PTP (0–200 µg/mL) for another 24 h. Next, 10 µL of CCK-8 solution was added, and the cells were subsequently cultured for 1.5 h. Absorbance at 450 nm was measured using a Varioskan Flash multimode microplate reader (Thermo Fisher Scientific, Waltham, MA, USA). The cytocompatibility of PTP was also assessed in C6 and BEAS-2B cells using the same method.

To assess the preliminary safety in vivo, five BALB/c nude mice were treated with PTP at a single dose (1 mg, 100 µL), and another five mice treated with saline were used as the control group. The body weight, physical activity, and death of each mouse were recorded during the following seven days. The major organs of the mice, including the heart, liver, spleen, lung, and kidneys, were harvested for histological examination.

### 2.4. Western Blot Assay and Immunofluorescence Staining

Plectin expression in BxPC-3 cells was confirmed using western blotting. PANC-1 and β-TC-6 cells were set as the positive and negative controls, respectively. Total proteins from BxPC-3, PANC-1, and β-TC-6 cells were extracted using RIPA buffer supplemented with PMSF and quantified using a BCA protein assay kit. The protein samples were separated by a 10% SDS-PAGE and transferred onto polyvinylidene fluoride membranes. After blocking with 5% non-fat milk in TBST for 1 h, the bands were incubated overnight at 4 °C with the primary antibody at a concentration of 1:1000 (anti-glucose transporter plectin antibody and tubulin used as endogenous controls). After washing three times in TBST, the membranes were incubated with goat anti-rabbit secondary antibodies at a concentration of 1:1000 for 1.5 h at room temperature. After washing in TBST, the protein bands were identified using ECL reagents. The blots were quantified using the ImageJ software.

To compare the expression of plectin protein in selected tumor cells (U87, C6, A549, 4T1, and BxPC-3), immunofluorescence staining assays were performed. BEAS-2B cells were used as the negative control. Briefly, tumor cells (2 × 10^5^) were seeded into glass-bottom dishes in 2 mL of medium. After 24 h incubation, the cells were fixed with 4% paraformaldehyde for 20 min, blocked with 5% bovine serum albumin containing 1% Triton X-100 for 30 min, and incubated with a primary antibody against plectin (1:400) (rabbit IgG, Cell Signaling Technology, USA) overnight at 4 °C. The cells were then incubated with anti-rabbit Alexa Fluor 597 IgG (YEASEN, Shanghai, China) as a secondary antibody (1:200) for 1 h at room temperature. Finally, nuclei were counterstained with 4′6-diamidino-2-phenylindole (DAPI) for 5 min. Photographic images were obtained and analyzed using a Leica SP8 laser confocal microscope (Wetzlar, Germany).

### 2.5. Flow Cytometry Analysis and Confocal Microscopy

The ability of PTP to target tumor cells in vitro was evaluated by flow cytometry. Briefly, taking C6 cells as an example, cells were seeded into 6-well plates at a density of 2 × 10^5^ cells/well. After 24 h, the medium was replaced with 1 mL of fresh serum-free medium containing FITC-PTP at different concentrations (0–40 μM). After another 4 h, the cells were washed, trypsinized, resuspended in 200 µL PBS containing 2% FBS, and measured using BD AccuriTM C6 Flow Cytometer (BD Biosciences, USA). A minimum of 10,000 events were recorded for each sample. The targeting abilities of PTP towards BEAS-2B (negative control), U87, 4T1, A549, BxPC-3, and PANC-1 (positive control) in vitro were evaluated according to a similar method.

The specificity of PTP towards tumor cells (U87, C6, 4T1, A549, and BxPC-3) in vitro was also tested by confocal microscopy. BEAS-2B cells were used as the negative control. Briefly, taking BxPC-3 cells as an example, the cells were seeded at a 5 × 10^4^ cells/mL density in glass-bottom cell culture dishes. The cells were incubated in a medium containing 10 μM FITC-PTP for 4 h. The cells were then rinsed, fixed, counterstained, and observed under a fluorescence microscope (Leica SP8, Wetzlar, Germany).

### 2.6. Ex Vivo Fluorescent Imaging

Fluorescent imaging was performed in five types of tumor-bearing mice (U87, C6, 4T1, A549, and BxPC-3) to test the in vivo targeting ability of FITC-PTP. Tumor-bearing mice were intravenously injected with a PBS solution of FITC-PTP (150 µL, 1 mg/mL), and one mouse was injected with PBS as a control. Considering the limited penetration of FITC, tumor-bearing mice were sacrificed to harvest the main organs and tumors at 1 h post-injection for ex vivo fluorescent imaging. Fluorescent images were acquired using IVIS Spectrum optical imaging (IVIS Lumina Series III, PerkinElmer, Waltham, MA, USA) with excitation at 535 nm and emission at 580 nm.

### 2.7. Preparation of [^99m^Tc]Tc-HYNIC-PTP

PTP modified with HYNIC was radiolabeled with [^99m^Tc]Tc according to a previously published procedure [36]. In brief, 10–200 µL of HYNIC-PTP solution (1 mg/mL in water), 0.5 mL EDDA solution (20 mg/mL in 0.1 M NaOH), 0.5 mL tricine solution (40 mg/mL in 0.2 M PBS, pH = 6.0), 1 mL Na[^99m^Tc]TcO_4_ solution (50 mCi/mL), and 50 µL SnCl_2_ solution (1 mg/mL in 0.1 M HCl) were mixed and heated in a boiling water bath for 15 min. After cooling to room temperature, the final solution was analyzed using instant thin-layer chromatography (ITLC) and radio-high-performance liquid chromatography (radio-HPLC).

### 2.8. Quality Control

The [^99m^Tc]Tc-labeled PTP was characterized using an Agilent 1260 HPLC system (Agilent Technologies, Santa Clara, CA, USA) equipped with a UV-vis detector (λ = 220 nm) and a radioactive flow detector (BioScan, Poway, CA, USA). A SunFire C18 column (5 µm, 4.6 × 250 mm, Waters, Japan) was used at a 1 mL/min flow rate with the following gradient method: 0.1% trifluoroacetic acid in H_2_O and CH_3_CN (0–20 min, 15–45% CH_3_CN). The RCP of [^99m^Tc]Tc-HYNIC-PTP was determined by radio-HPLC and could also be rapidly analyzed by ITLC in a system consisting of silica gel 60 F254 TLC plates (Merck, Germany) and 50% acetonitrile as mobile phase. To assess the stability in vitro, the formed [^99m^Tc]Tc-HYNIC-PTP (500 μL, 1 mCi) was mixed with 500 μL of PBS (0.1 M, pH = 7.4) and 500 µL of cysteine solution (100-fold molar excess over the PTP) at room temperature, and 500 µL of FBS at 37 °C. The RCPs were tested by ITLC at different time intervals (0–6 h).

### 2.9. Pharmacokinetics

The pharmacokinetic profile of [^99m^Tc]Tc-HYNIC-PTP was investigated in healthy ICR mice. The mice were randomly divided into nine groups (n = 3), and each mouse was intravenously injected with [^99m^Tc]Tc-HYNIC-PTP at a dose of 20 µCi in a 200 µL solution. One hundred microliters of blood from each mouse was immediately collected and weighed at designated times (1, 2, 5, 15, 30, 60, 120, 240, and 360 min), and radioactivity was measured using a γ-counter (CAPINTEC, USA) to calculate the percent uptake of the injected dose per gram (%ID/g). In addition, the pharmacokinetic data were analyzed by DAS 2.0 (Shanghai, China) using a two-compartment model to calculate the half-life of [^99m^Tc]Tc-HYNIC-PTP in the blood.

### 2.10. SPECT Imaging In Vivo and Immunohistochemistry Assays

The feasibility of [^99m^Tc]Tc-HYNIC-PTP as a SPECT probe for tumor imaging was examined using established subcutaneous tumor models. Briefly, [^99m^Tc]Tc-HYNIC-PTP solution (200 µL, [^99m^Tc]Tc = 10 mCi/mL, corresponding to 2 µg of HYNIC-PTP) was intravenously injected into tumor-bearing mice. SPECT images were acquired 0.5, 1, 2, and 4 h after injection using a SPECT imaging system equipped with a Xeleris 2.0 workstation and low-energy general-purpose collimators (Infinia, Denver, CO, USA). After SPECT imaging, the tumors and muscle tissues were excised to confirm plectin expression levels by immunohistochemistry. Briefly, excised specimens were fixed in formalin and embedded in paraffin. The sections (5 μm) were subjected to heat treatment in a citrate solution for antigen retrieval and blocking with 3% bovine serum albumin for 30 min. Then the sections were incubated with anti-plectin primary antibody (1:200) overnight at 4 °C. After incubation with secondary antibody (K5007, DAKO, Carpinteria, CA, USA) at room temperature for 50 min, the slides were treated with 1:100 DAB (K5007, DAKO, USA) and counterstained with hematoxylin. At least three sections of each specimen were chosen, and 5–10 high-power visual fields were randomly selected to calculate the positive area of staining. The average areas were then used to plot histograms.

### 2.11. Biodistribution

Mice bearing C6 or 4T1 tumor xenografts were intravenously injected with [^99m^Tc]Tc-HYNIC-PTP at a dose of 20 µCi in 200 µL solution (corresponding to 20 ng of HYNIC-PTP) to evaluate biodistribution in tumors and major organs (n = 3). The mice were sacrificed 0.5, 1, 2, and 4 h after injection to collect blood and tissue samples, including the heart, lung, liver, stomach, intestine, spleen, kidneys, muscle, and tumor. These samples were immediately weighed, and their radioactivity was measured using a γ-counter. The amount of radioactivity is expressed as %ID/g.

### 2.12. Statistical Analysis

Data are presented as the mean ± standard deviation, and one-way analysis of variance was performed to evaluate the significance of the data. A *p*-value of 0.05 was selected as the threshold of significance, and the data were denoted with (*) for *p* < 0.05, (**) for *p* < 0.01 and (***) for *p* < 0.001.

## 3. Results

### 3.1. Preliminary Toxicity Assessment

Biocompatibility of the PTP peptide was investigated in different cell lines in vitro and in normal nude mice in vivo. The results showed that the cell viabilities after treatment with PTP at the studied concentrations remained more than 90% at 24 h (Appendix A), suggesting excellent cytocompatibility in vitro. None of the mice died in the toxicity study, and no significant difference in body weight and physical activity of mice treated with PTP was found compared with the saline group within seven days. Moreover, biosafety after treatment with PTP was checked by observing the H&E staining of the main organs, including the heart, liver, lung, spleen, and kidneys (Appendix A). Unsurprisingly, no significant difference in tissue damage, necrotic areas, or abnormalities was found between the PTP and saline groups, suggesting the excellent biocompatibility of PTP in vivo.

### 3.2. Plectin Expression in Tumor Cells

We examined the expression of plectin in BxPC-3 cells using western blotting. PANC-1 and β-TC-6 cells were used as the positive and negative controls, respectively. As presented in Figure 1A,B, PANC-1 and BxPC-3 cells exhibited high plectin expression levels with no statistical difference, while no obvious plectin was expressed in β-TC-6 cells, consistent with the literature [30]. Furthermore, the plectin expression levels in different tumor cells (U87, C6, A549, 4T1, and BxPC-3) were compared with those in BEAS-2B cells (negative control) using immunofluorescence staining assays. The results revealed strong fluorescence signals in the U87, C6, A549, 4T1, and BxPC-3 cells compared with BEAS-2B cells, suggesting plectin overexpression in these tumor cells. Quantitative analysis showed that plectin expression levels in the selected tumor cells followed the order from lowest to highest: BEAS-2B, U87, C6, A549, 4T1, and BxPC-3.

### 3.3. Specificity of PTP to Tumor Cells In Vitro

The ability of the PTP peptide to target different tumor cells in vitro was examined by flow cytometry. We first performed a series of flow cytometry assays using C6 cells at different FITC-PTP concentrations (0.5–40 μM) to confirm the appropriate concentration. As shown in Figure 2A and Appendix A, the fluorescence intensity of C6 cells after 4 h of incubation gradually increased with the FITC-PTP concentration, and no changes in the fluorescence intensities could be found between the PBS group and FITC-PTP at relatively low concentrations of 0.5 and 1 μM. When the concentration ranged from 2–40 μM, the fluorescence intensity was almost proportional to the concentration of FITC-PTP. Based on these results, the concentration of FITC-PTP used in flow cytometry for other tumor cells was 10 μM. As shown in Figure 2B, the fluorescence intensities in tumor cells after 4 h of incubation with FITC-PTP were significantly higher than those of the negative control cell (BEAS-2B). Notably, after quantitative analysis, the fluorescence intensities followed the same order as their plectin expression levels in tumor cells, and BxPC-3 cells showed the highest FITC signal intensities. Similarly, confocal microscopy also showed stronger FITC fluorescence intensities in tumor cells than in BEAS-2B cells (Figure 3). Together, these data from flow cytometry and confocal microscopy demonstrated the high specificity of the PTP peptide for tumor cells.

### 3.4. Ex Vivo Fluorescent Imaging

The specificity of PTP to the studied tumor cells was investigated in vivo using fluorescence imaging. Five types of tumor-bearing mice (U87, C6, A549, 4T1, and BxPC-3) were intravenously injected with FITC-PTP, and the major organs and tissues, including the heart, liver, lung, spleen, kidneys, tumor, and muscle, were collected 1 h after injection. As shown in Figure 4, a high accumulation of FITC-PTP was found in tumors, but low fluorescence intensities were observed in major organs, such as the liver and kidneys. The preferential accumulation in the tumor made a clear distinction from other tissues, suggesting the good specificity of PTP for tumor imaging. These data further verified that PTP could specifically deliver imaging agents to tumors, making it a promising agent for tumor diagnosis.

### 3.5. Radiochemistry

PTP could be effectively radiolabeled with [^99m^Tc]Tc via the bifunctional chelator HYNIC, using tricine and EDDA as co-ligands. The RCPs of [^99m^Tc]Tc-HYNIC-PTP were analyzed by radio-HPLC and ITLC. As the radio-HPLC results show in Figure 5A, a single radioactive peak of [^99m^Tc]Tc-HYNIC-PTP was observed with a retention time of 9.71 min, matching with the corresponding HYNIC-PTP (8.92 min), while the retention time of Na[^99m^Tc]TcO_4_ was 3.13 min. The data from ITLC showed that the colloidal [^99m^Tc]Tc and Na[^99m^Tc]TcO_4_ had retention factors (R_f_) of 0–0.2 and 0.8–1.0, respectively, while the [^99m^Tc]Tc-HYNIC-PTP displayed an R_f_ of 0.4–0.6 (Figure 5B). To investigate the optimal labeling conditions, a series of experiments were performed using different doses of HYNIC-PTP for radiolabeling. Although the RCPs were all over 90% in the range of 10–200 μg, the RCP could be more than 95% when the dose of HYNIC-PTP for radiolabeling was above 50 μg (Figure 5C), suggesting no need for further purification procedures. Therefore, [^99m^Tc]Tc-HYNIC-PTP used in subsequent experiments was prepared under the optimum reaction conditions of 50 μg HYNIC-PTP, 50 mCi Na[^99m^Tc]TcO_4_, and 50 μg SnCl_2_, with a specific radioactivity of more than 1000 Ci/g. Furthermore, no evident changes in RCPs were found in PBS in the presence of a 100-fold molar excess of cysteine at room temperature and FBS at 37 °C within 6 h, suggesting high stability of [^99m^Tc]Tc-HYNIC-PTP in vitro (Figure 5D).

### 3.6. Pharmacokinetics

The radioactivity-time curve is shown in Appendix A. At 1 min post-injection, the radioactivity in blood was 14.52%ID/g, which rapidly decreased to 1.35%ID/g at 30 min post-injection. Less than 0.01%ID/g could be recovered from the blood pool at 360 min post-injection. The distribution-phase half-life (t_1/2_ alpha) and clear-phase half-life (t_1/2_ beta) of [^99m^Tc]Tc-HYNIC-PTP were estimated to be 0.88 and 9.17 min, respectively.

### 3.7. Targeted SPECT Imaging of Tumors In Vivo and Immunohistochemistry Assays

To evaluate the feasibility of [^99m^Tc]Tc-HYNIC-PTP as a probe for tumor detection in vivo, SPECT imaging was conducted in five types of tumor-bearing mice (U87, C6, A549, 4T1, and BxPC-3). As shown in Figure 6A, the probe exhibited a similar distribution pattern in these mice. The main radioactivity was found in the kidneys and bladder, with low uptake in the heart, lung, liver, spleen, intestines, and muscle, suggesting that [^99m^Tc]Tc-HYNIC-PTP was predominantly cleared through the urinary system. Notably, fast clearance was observed in the blood and lungs, with background radioactivity levels remaining at 2 h post-injection. Conversely, evident uptake in the tumors was observed during the study period. The tumor accumulation of [^99m^Tc]Tc-HYNIC-PTP was observed at 0.5 h post-injection and sustained with time. The tumor-to-muscle (T/M) SPECT signal ratios at different time points were measured to determine the optimal time point for SPECT imaging (Figure 6B). Although satisfactory tumor conspicuity was detected in these five tumor types at 1 h post-injection, the highest T/M ratios occurred at different time points. The highest T/M ratios in U87 and C6 tumors were observed at 2 h post-injection, while the T/M ratios in A549, 4T1, and BxPC-3 continued to increase during the study period. After SPECT imaging, the plectin expression levels in tumors were confirmed by immunohistochemistry to analyze the correlation with tumor uptake, and muscle tissues were set as the negative control. As shown in Figure 6C and Appendix A, the levels of plectin expression showed high consistency between immunohistochemistry and immunofluorescence staining. Tumors with higher plectin expression levels had better T/M ratios, indicating a higher tumor uptake. This correlation further validated the specificity of [^99m^Tc]Tc-HYNIC-PTP for the plectin receptor in vivo.

### 3.8. Biodistribution

A biodistribution experiment was performed in C6 and 4T1 tumors to validate the correlation between tumor uptake and plectin expression. Similar to the SPECT results, the major radioactivity accumulated in the kidneys, with relatively low accumulation in other organs and rapid clearance from the blood and lungs (Figure 7). Importantly, as expected, tumors with higher expression levels of plectin possessed better tumor uptake and T/M ratios. For example, the tumor uptake and T/M ratio in C6 tumors at 1 h post-injection were 0.22 ± 0.02%ID/g and 2.78 ± 0.25, lower than those of 4T1 tumors (0.45 ± 0.02%ID/g and 4.93 ± 0.48). Although both the tumor uptake in C6 and 4T1 tumors decreased at 4 h post-injection (0.13 ± 0.01%ID/g and 0.25 ± 0.03%ID/g), a higher T/M ratio was displayed in 4T1 tumors when compared with that in C6 tumors at the same time point (3.37 ± 0.57 vs. 5.05 ± 0.97).

## 4. Discussion

The design and development of molecular imaging probes for nuclear medicine have attracted significant interest. As a result, an increasing number of receptors have been explored in this field, such as fibroblast activation proteins, prostate-specific membrane antigens, and somatostatin receptors [2]. In addition, several types of targeting molecules have been investigated for their receptors [4,5,6,7,8,9]. Because of their attractive advantages in constructing molecular imaging probes, including high affinity and specificity, fast blood clearance, low immunogenicity, and easy modification, many peptides have been labeled with radionuclides for tumor receptor imaging, and some are being widely used in clinics [12]. Given these successes, peptide-based probes have become established strategies in molecular imaging, further prompting the discovery and development of novel peptide-based imaging agents for disease diagnosis [11].

In this study, the heptapeptide PTP was selected as a targeting molecule to develop a new peptide-based probe for tumor imaging because of its high binding affinity and specificity for the plectin receptor, which is widely overexpressed in various tumors and has great potential as a biomarker for tumor detection. We first tested the cytocompatibility in vitro and conducted a preliminary toxicity assessment of PTP peptides in vivo. As expected, no remarkable cytotoxicity to the tumor or normal cells and no abnormality in normal mice were detected, indicating the high safety of the peptide. Although many reports have proven the specificity of the PTP peptide for plectin, most studies have focused on pancreatic cancer. Hence, the targeting abilities of PTP to tumor cells (U87, C6, A549, 4T1, and BxPC-3) used in the present study were evaluated in vitro. In the flow cytometry study, FITC-PTP exhibited significantly higher fluorescence signals in these plectin-overexpressing tumor cells than in BEAS-2B cells (negative control), and the fluorescence intensities showed good correlations with their plectin expression results from immunofluorescent staining. The fluorescent signal in BEAS-2B cells suggested the physiological expression of plectin in normal lung cells. Notably, BxPC-3 and PANC-1 cells (positive control) had the highest FITC signal intensities, indicating similar plectin expression levels, confirmed by western blotting. The specificity of the PTP peptide to tumor cells was also verified by confocal microscopy. According to the flow cytometry study results, compared to BEAS-2B cells, tumor cells displayed stronger FITC fluorescence intensities. Moreover, tumor targeting by FITC-PTP was demonstrated by ex vivo fluorescent imaging. The high accumulation of FITC-PTP in tumors and the low fluorescence signals in other tissues revealed satisfactory specificity and selectivity for tumor detection in vivo.

To provide an easy method for the [^99m^Tc]Tc-labeled PTP preparation, HYNIC was attached to the N-terminus of the peptide, and HYNIC-PTP was labeled with [^99m^Tc]Tc using tricine and EDDA as co-ligands. The main reason for using HYNIC as a bifunctional chelator was the facile conjugation to the peptide with high RCP in [^99m^Tc]Tc radiolabeling. Using EDDA and tricine as co-ligands could improve the hydrophilicity and pharmacokinetics of [^99m^Tc]Tc-labeled PTP for rapid elimination through the urinary system, which was demonstrated in a subsequent SPECT imaging study. Hence, [^99m^Tc]Tc-HYNIC-PTP could be acquired in 15 min with high RCP and specific radioactivity in this study. This method was easy to perform because the preparation was achieved in one pot, and the RCP was greater than 90% without further purification, even in 10 μg HYNIC-PTP. The convenience of radiolabeling makes the kit formulation available. Moreover, [^99m^Tc]Tc-HYNIC-PTP displayed favorable stability in vitro in PBS, FBS, and cysteine solution for 6 h, supporting further investigations in animal models for tumor-targeting SPECT imaging.

SPECT images of mice injected with [^99m^Tc]Tc-HYNIC-PTP displayed fast accumulation and relatively high uptake in the tumors. The accumulation of [^99m^Tc]Tc-HYNIC-PTP in the five types of tumor-bearing mice was observable 0.5 h post-injection and maintained during the study period. However, these mice’s T/M ratios at different time points showed inconsistent trends, probably related to the tumor plectin status. The highest T/M ratios of tumors with relatively low plectin expression, such as U87 and C6, were observed at 1 and 2 h after injection, respectively. In contrast, the T/M ratios continuously increased in the tumors with higher plectin expression levels (4T1, A549, and BxPC-3). SPECT imaging was in accordance with the immunohistochemistry results, indicating good specificity of PTP to different tumors in vivo and enhanced retention of [^99m^Tc]Tc-HYNIC-PTP in tumors with higher plectin expression levels. The biodistribution experiment further supported this, which showed that 4T1 tumors exhibited better tumor uptake and T/M ratio at each time point than C6 tumors. Additionally, the biodistribution and pharmacokinetic data revealed that [^99m^Tc]Tc-HYNIC-PTP had a rapid elimination from the blood and muscle but a relatively slow clearance in tumors, which allowed sufficient conspicuity of tumor imaging within 2 h after injection. Furthermore, except for the kidneys and bladder, the accumulation of radioactivity in other organs was low. However, rapid clearance of [^99m^Tc]Tc-HYNIC-PTP was still found in some normal organs, such as the lungs, which was beneficial in avoiding unnecessary radiation dose burden. Notably, both [^99m^Tc]Tc-HYNIC-PTP and FITC-PTP showed distinct tumor uptake, while different biodistribution in liver and kidneys due to their structural differences. The hydrophilicity of HYNIC led to a high accumulation of [^99m^Tc]Tc-HYNIC-PTP in kidneys. Still, FITC-PTP had an obvious fluorescence signal in the liver, which was in line with the literature [30]. This suggested that the introduction of FITC or HYNIC probably changed the biodistribution and pharmacokinetics of PTP, with no significant influence on targeting ability.

The PTP peptide has been utilized as a targeting molecule to enhance the specificity of imaging agents and the efficacy of drug delivery systems in various studies, such as PTP-combined RGD peptides as a bispecific molecular probe for pancreatic cancer imaging and PTP-modified nanopolyplexes for pancreatic cancer therapy [30,31]. Although nanoplatform-based probes have many unique characteristics for tumor imaging, they often suffer from high accumulation in the reticuloendothelial system and undesired toxicological risks. Therefore, the development of radionuclide-labeled PTP shows great advantages because of the probe’s high sensitivity and tracer amount, which is beneficial for clinical translation. However, exploration of the utility of PTP-based radiopharmaceuticals for tumor diagnosis is still lacking. Only one previous study reported [^111^In]In-labeled tetrameric PTP peptide ([^111^In]In-tPTP) as a SPECT agent for pancreatic cancer imaging [18]. In that study, [^111^In]In-tPTP showed good specificity and selectivity in distinguishing pancreatic cancer from its metastases in an orthotopic and liver metastasis mouse model. Nevertheless, the RCP of [^111^In]In-tPTP was unsatisfactory, and post-labeling purification was required. Furthermore, the low availability and high production cost of [^111^In]In radionuclides limit further clinical application. In contrast to [^111^In]In, [^99m^Tc]Tc is the most commonly used radionuclide for SPECT imaging in the clinic because of its excellent physicochemical properties and convenient acquisition from commercial generators at a low cost, making it suitable for peptide-based probes. This study designed and successfully synthesized [^99m^Tc]Tc-HYNIC-PTP to develop a plectin-targeting probe for tumor imaging. Plectin overexpression has been identified in a wide range of cancers because of its involvement in various cellular activities in tumors, such as cell proliferation, survival, migration, and invasion [15]. This motivated us to evaluate the feasibility of [^99m^Tc]Tc-HYNIC-PTP as a SPECT imaging agent for multiple cancers, not just pancreatic carcinoma. Although [^99m^Tc]Tc-HYNIC-PTP displayed satisfactory imaging performance in selected tumors, its radiation dosimetry was unclear, and its extended application for cancer imaging was also uncertain. Further studies are required to address these issues.

## 5. Conclusions

In summary, plectin has been proven to be overexpressed in various tumors, serving as a specific target for cancer diagnosis. In this study, we prepared a plectin-targeting imaging agent, [^99m^Tc]Tc-HYNIC-PTP using a simple method with high RCP and stability and evaluated its feasibility as a SPECT probe for tumor imaging. The data demonstrated a distinct accumulation of [^99m^Tc]Tc-HYNIC-PTP in five types of tumor-bearing mice with favorable biodistribution and pharmacokinetics, and tumor uptake correlated with their plectin expression levels. Although this probe holds great potential as a novel strategy for tumor imaging, further studies are needed to verify its validity in other plectin-overexpressing tumors and acquire sufficient preclinical data before clinical trials.

## Figures and Tables

**Figure 1 pharmaceutics-14-00996-f001:**
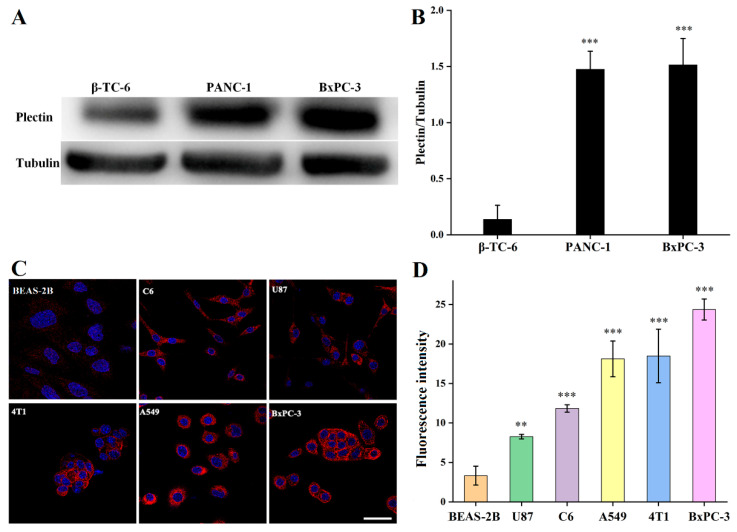
(**A**) Western blot results of plectin and (**B**) densitometry values in β-TC-6, PANC-1, and BxPC-3 cells. (**C**) Immunofluorescence staining of plectin and (**D**) quantitative analysis in different cell lines (BEAS-2B, BxPC-3, C6, U87, A549, and 4T1). The scale bar represents 50 µm for all panels. (**) for *p* < 0.01, (***) for *p* < 0.001.

**Figure 2 pharmaceutics-14-00996-f002:**
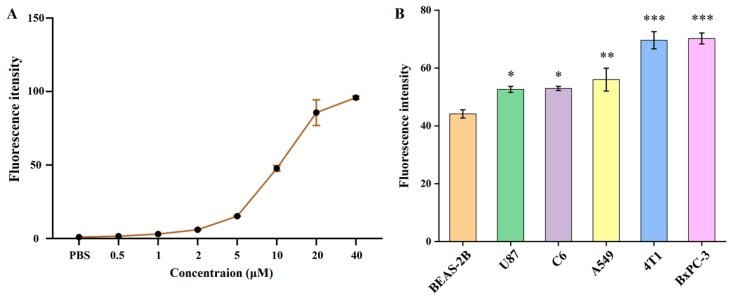
(**A**) Fluorescence intensities of C6 cells treated with FITC-PTP at different concentrations (0 to 40 μM) for 4 h, and (**B**) fluorescence intensities in six types of cells treated with FITC-PTP at 10 μM for 4 h. (*) for *p* < 0.5, (**) for *p* < 0.01, (***) for *p* < 0.001.

**Figure 3 pharmaceutics-14-00996-f003:**
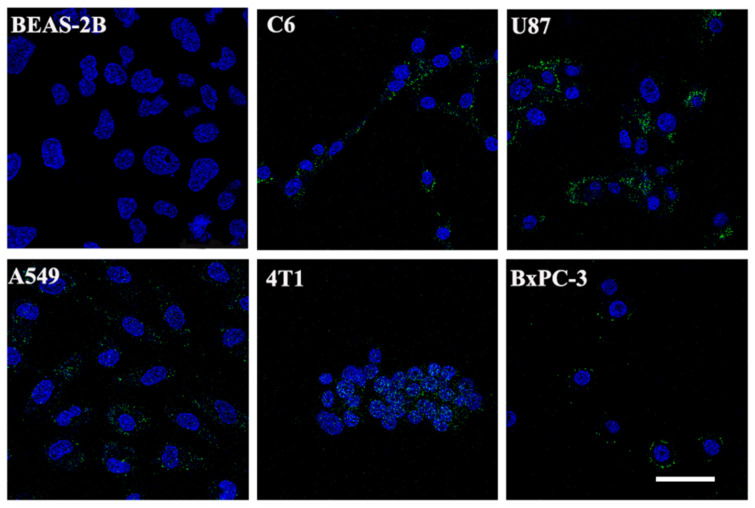
Confocal microscopic images of six types of cells treated with 10 μM FITC-PTP for 4 h. The scale bar represents 50 µm for all panels.

**Figure 4 pharmaceutics-14-00996-f004:**
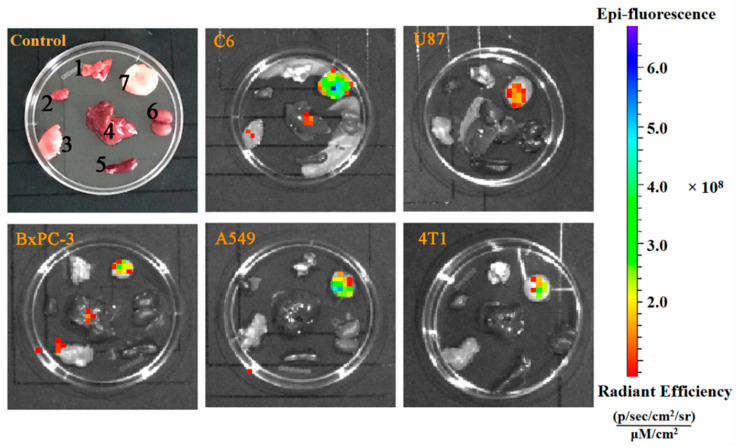
Ex vivo fluorescent images of tumor-bearing mice (U87, C6, A549, 4T1, and BxPC-3) at 1 h post-injection of FITC-PTP. C6 tumor-bearing mice treated with PBS were set as the control group. Samples were (1) lung, (2) heart, (3) muscle, (4) liver, (5) spleen, (6) kidneys, and (7) tumor, respectively.

**Figure 5 pharmaceutics-14-00996-f005:**
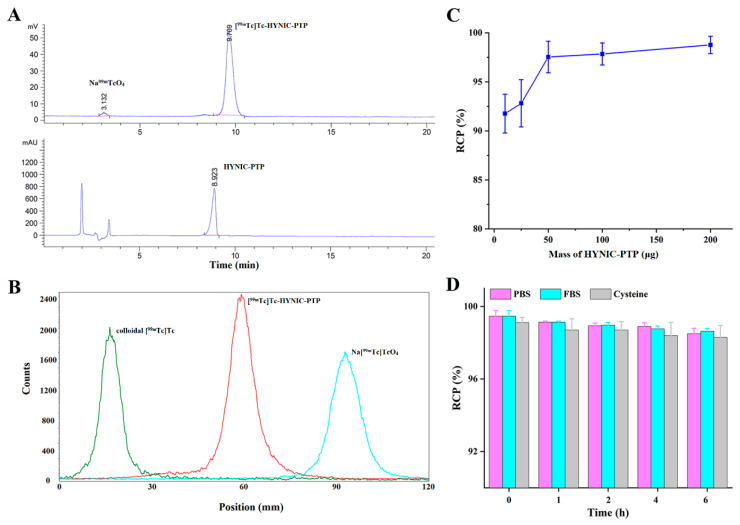
(**A**) Radio-HPLC and (**B**) ITLC results of [^99m^Tc]Tc-HYNIC-PTP. (**C**) RCPs of [^99m^Tc]Tc-HYNIC-PTP at different doses of HYNIC-PTP. (**D**) In vitro stability of [^99m^Tc]Tc-HYNIC-PTP in PBS, FBS, and cysteine solution at different time points.

**Figure 6 pharmaceutics-14-00996-f006:**
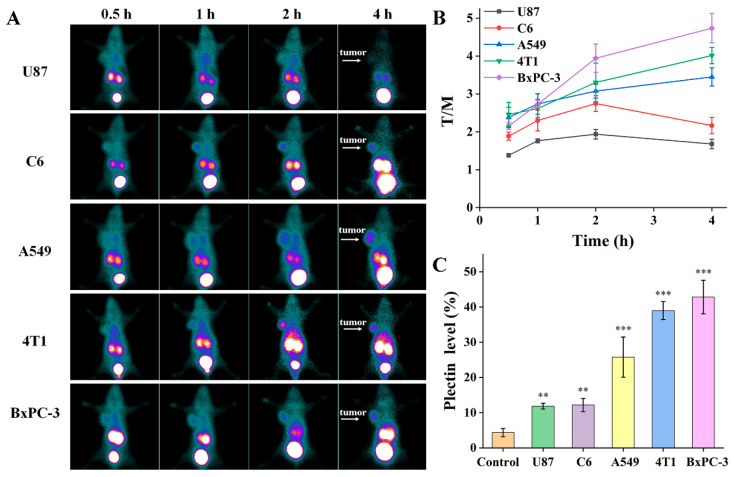
(**A**) SPECT images, (**B**) tumor-to-muscle (T/M) SPECT signal ratios of [^99m^Tc]Tc-HYNIC-PTP in five types of tumor-bearing mice at different time points, and (**C**) the relative plectin expression levels. (**) for *p* < 0.01, (***) for *p* < 0.001.

**Figure 7 pharmaceutics-14-00996-f007:**
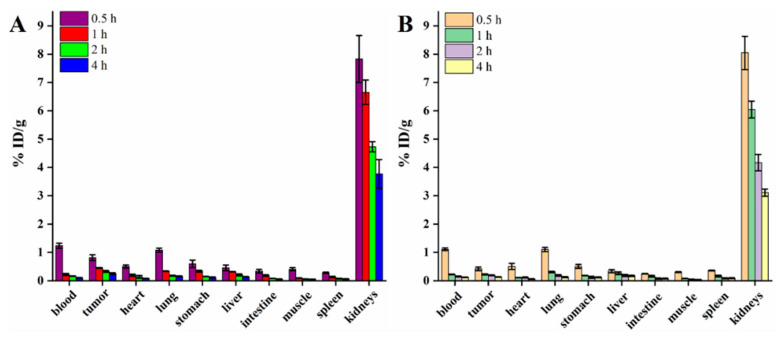
Biodistribution of [^99m^Tc]Tc-HYNIC-PTP in (**A**) 4T1 and (**B**) C6 tumor-bearing mice at different time points.

## Data Availability

All data are contained within the article or Appendix A.

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
