# Peer review of "[99mTc]Tc-Labeled Plectin-Targeting Peptide as a Novel SPECT Probe for Tumor Imaging"

_pharmaceutics, 2022, doi:10.3390/pharmaceutics14050996_

Round 1

Reviewer 1 Report

This is a very nice piece of work describingt he radiolabelling of a peptide targeting Plectin. The work is fairly novel with the only other radiolabelled derivative described being the In-111 tetrapeptide.

The work described presents novel information on the presence of Plectin in various tumour models extending what is known in the literature from earlier  work described in  pancreatic cancer models.

The  manuscript and the supplementary notes are well described and free of any grammatical/spelling mistakes.

The methodology, results and experimental are well written, comprehensive and easy to follow and review. As a preliminary study the 99mTc-Hynic-PTP is well characterised and further studies are warranted.

Two minor questions (more of a curiosity rather than a criticism) are:

 The authors used hynic- tricine as the chelators for Tc-99m labelling. However, it is well known that the complexing agent itself and even the co-ligand with hynic can influence the pharmacokinetics of the tracer. Have the authors looked at other 99mTc-chelators to see if that influences biodistribution, tumour uptake, blood clearance etc., and hence get a better tumour to blood ratio?

The need for quick imaging post tracer administration is appreciated and a goal for the investigators. However, the tumour to blood ratio is a little low even at 4 h. Have the authors looked at longer time points.

Author Response

Reviewer 1:

  1. The authors used hynic- tricine as the chelators for Tc-99m labelling. However, it is well known that the complexing agent itself and even the co-ligand with hynic can influence the pharmacokinetics of the tracer. Have the authors looked at other 99mTc-chelators to see if that influences biodistribution, tumour uptake, blood clearance etc., and hence get a better tumour to blood ratio?

Author reply: We thank the reviewer for her/his great comments and questions. The introduction of bifunctional chelating agents probably changes the biodistribution and pharmacokinetics of small peptides. In our previous studies, several peptides were radiolabeled with 99mTc through via tricarbonyl chemistry, and obvious radioactivity accumulation was found in the liver and kidneys in SEPCT imaging (Contrast Media Mol Imaging, 2019;2019:9502712. New J Chem, 2020;44:14947-14952). In addition to HYNIC, we also tried DTPA as a chelator to conjugate PTP for 99mTc radiolabeling, but the radiochemical yield was less than 90%, suggesting an additional purification process. Therefore, the main reason for using HYNIC in this study was the facile conjugation to the PTP with high radiochemical purity in 99mTc radiolabeling. The use of EDDA and tricine as co-ligands could improve the hydrophilicity and pharmacokinetics of 99mTc-labeled PTP for rapid elimination through the urinary system. We agree with the comment regarding the possible influence on SPECT imaging using other 99mTc-chelators. This is a worthy issue to investigate, and we will carry out in the follow-up work.

  1. The need for quick imaging post tracer administration is appreciated and a goal for the investigators. However, the tumour to blood ratio is a little low even at 4 h. Have the authors looked at longer time points.

Author reply: We appreciate the reviewer for her/his good comments. We first evaluated the SPECT imaging of 99mTc-HYNIC-PTP within 6 hours in C6 and BxPC-3 tumor-bearing mice. As excepted, higher T/M ratio was found in BxPC-3 tumors with high plectin expression at 6 h post-injection when compared to that at 4 h. Inversely, C6 tumors with low plectin expression had a lower T/M ratio at 6 h post-injection. However, the results of SPECT imaging at 6 h were similar to those at 4 h. Therefore, we performed SPECT imaging within 4 hours for other selected tumors in this study.

Reviewer 2 Report

The manuscript by Gong et al. describes the development of a novel plectin-targeting peptide, labelled with Tc-99m for SPECT application.

The manuscript is well written and the authors presents a comprehensive spectrum of in vitro and in vivo data using cell lines having different plectin level expression. Nevertheless, several points have to be addressed prior publication.

The authors should explain the structural differences between the two conjugates used for in vitro (FITC-PTP) and in vivo (99mTc- labelled Hynic-PTP) and how these differences can impact on their affinities towards the targeted receptor. Moreover, no affinity data are reported, neither in vivo blocking studies are performed to clearly state that this new conjugate can recognize and bind selectively to the receptor. Even though the authors reported significant differences among positive and negative cells in the flow cytometry study, a fluorescent signal of 40% in the negative control cannot be considered negligible. The authors should explain how they justify the lack of uptake in the kidney in the ex vivo fluorescent imaging study as compared as the high accumulation of the radiotracer. The tumor uptake is extremely low, fast washed out and without blocking study (specific uptake?!) cannot be carefully judged. Did the author perform study on in vivo stability and in vitro radioligand assays to confirm binding properties and specificity of their Hynic-conjugate?

Minor:

Page 2 line 80: SEPCT should be changed in SPECT

Page 11 line 16…: please, add reference to this sentence.

Author Response

Reviewer 2:

  1. The authors should explain the structural differences between the two conjugates used for in vitro (FITC-PTP) and in vivo (99mTc- labelled Hynic-PTP) and how these differences can impact on their affinities towards the targeted receptor. Moreover, no affinity data are reported, neither in vivo blocking studies are performed to clearly state that this new conjugate can recognize and bind selectively to the receptor. Even though the authors reported significant differences among positive and negative cells in the flow cytometry study, a fluorescent signal of 40% in the negative control cannot be considered negligible.

Author reply: We appreciate the reviewer for her/his professional comments. The N-terminal of PTP was respectively modified with HYNIC and FITC to form HYNIC-PTP and FITC-PTP, which was used for the 99mTc radiolabeling or the evaluation of targeting ability by flow cytometry and confocal imaging. The introduction of FITC or HYNIC probably changes the biodistribution and pharmacokinetics of PTP, but no significant influence on targeting ability. The specificity and affinity of PTP for plectin have been confirmed in the literature (PLoS Med, 2008;5(4):e85. Biomaterials, 2018;183:173-184). Moreover, a 111In-labeled tetrameric PTP peptide was reported as a SPECT agent for pancreatic cancer imaging (Clin Cancer Res. 2011;17(2):302-9). Therefore, the main purpose of this study was to evaluate the feasibility of 99mTc-labeled PTP for SPECT imaging in tumors with different levels of plectin expression. We also performed the in vivo blocking experiments, and got an interesting finding. Co-injection of PTP and 99mTc-HYNIC-PTP within a certain range seemed to decrease nonspecific accumulation in lung and kidney and enhance T/M ratio. This issue is still being investigated, and we plan to report the results in our further work. In flow cytometry study, normal human lung epithelial BEAS-2B cells were selected as a control group to confirm the overexpression of plectin in tumor cells. The fluorescent signal of 40% also indicated the physiological expression of plectin in normal lung cells, which was consistent with the obvious accumulation of 99mTc-HYNIC-PTP in lung in the subsequent SPECT imaging and biodistribution experiments. According to the suggestions and comments, some description was added in the revised manuscript.

  1. The authors should explain how they justify the lack of uptake in the kidney in the ex vivo fluorescent imaging study as compared as the high accumulation of the radiotracer.

Author reply: We thank the reviewer for her/his valuable suggestions. As described above, the introduction of FITC or HYNIC probably changes the biodistribution and pharmacokinetics of PTP, but no significant influence on targeting ability. Both FITC-PTP and 99mTc-HYNIC-PTP showed distinct tumor uptake, while different bio distribution in liver and kidneys. The hydrophilicity of HYNIC leaded to high accumulation of the radiotracer in kidneys, and FITC-PTP showed accumulation in liver, which was in line with the literature (Biomaterials, 2018;183:173-184).

  1. The tumor uptake is extremely low, fast washed out and without blocking study (specific uptake?!) cannot be carefully judged. Did the author perform study on in vivo stability and in vitro radioligand assays to confirm binding properties and specificity of their Hynic-conjugate?

Author reply: We thank the reviewer for her/his good comments. As mentioned above, the specificity of PTP to plectin has been confirmed by several researchers (PLoS Med, 2008;5(4):e85. Biomaterials, 2018;183:173-184). The primary aim of this study was to evaluate the feasibility of 99mTc-labeled PTP for SPECT imaging in tumors with different levels of plectin expression. Although the tumor uptake was relatively low, the rapid clearance of 99mTc-HYNIC-PTP from blood and non-target tissues made acceptable T/M ratios in tumors with high plectin experssion. Moreover, the 111In-labeled tetrameric PTP peptide showed better tumor uptake than 99mTc-HYNIC-PTP, suggesting that the targeting ability of PTP can be further improved by appropriate modification, which is one of our future research directions.

Minor:

  1. Page 2 line 80: SEPCT should be changed in SPECT

Author reply: We thank the reviewer for her/his good comments. “SEPCT” has been changed to “SPECT” in the revised manuscript.

  1. Page 11 line 16…: please, add reference to this sentence.

Author reply: We thank the reviewer for her/his valuable suggestions. We have added the cited reference to this sentence.

Reviewer 3 Report

In the attached file.

Author Response

Reviewer 3:

Specific comments:

  1. Line 75: 99mTc-HYNIC-PTP - The nomenclature should follow “Consensus nomenclature rules for radiopharmaceutical chemistry — Setting the record straight” (Nucl Med Biol. 2017 Dec;55:v-xi). Correct way would be [99mTc]Tc-HYNIC-PTP. This corrections should be made throughout the whole manuscript.

Author reply: We thank the reviewer for her/his valuable comments. 99mTc-HYNIC-PTP has been corrected into [99mTc]Tc-HYNIC-PTP throughout the whole manuscript.

2.Line 80: “...novel SEPCT probe...” – should be “...novel SPECT probe...”

Author reply: We thank the reviewer for her/his good advice. In Line 80, “novel SEPCT probe” has been changed into “novel SPECT probe”.

  1. Line 234: the amount (mass) or molar of HYNIC-PTP injected to the animals is missing. Is this mass low enough not to saturate the receptors? Did you check that? The rationale for the mass (moles) used for animal studies has to be explained.

Author reply: We thank the reviewer for her/his professional suggestions. As described in Line 331, 99mTc-HYNIC-PTP used in subsequent experiments (including SPECT imaging and biodistribution) was prepared under the optimum reaction conditions of 50 μg HYNIC-PTP, 50 mCi Na99mTcO4, and 50 μg SnCl2, with a specific radioactivity of 1000 Ci/g. The dose of 99mTc-HYNIC-PTP injected to the animals was 2 mCi (200 μL, [99mTc] = 10 mCi/mL), corresponding to 2 μg of HYNIC-PTP that was significantly lower than the 20 μg of tPTP reported in the literature (Clin Cancer Res. 2011;17(2):302-9). Therefore, the mass used for animal studies was low enough. According to the suggestions and comments, the mass of HYNIC-PTP injected to the animals was added in the revised manuscript.

  1. Figure 5: Graph C – dose indicates activity. Should be changed to mass. The mass seems very high, up to 200 ug. It is not clear why you went so high with the mass. In principle you should seek high specific activity, which seems not to be the case here. Please explain, correct.

Author reply: We thank the reviewer for her/his valuable comments. The “dose” has been changed into “mass” in the revised manuscript. Several experiments were performed using different amount of HYNIC-PTP (10-200 ug) for radiolabeling to investigate the optimal labeling conditions. The 99mTc-HYNIC-PTP used in SPECT imaging was prepared under the optimum reaction conditions of 50 μg HYNIC-PTP, 50 mCi Na99mTcO4, and 50 μg SnCl2, with a specific radioactivity of 1000 Ci/g.

  1. Figure 6: Part A with SPECT images is very unclear. D part could potentially go into supplementary material.

Author reply: We thank the reviewer for her/his helpful comments. We have improved the quality of Part A. Part D of Figure 6 has been removed to supplementary material.

  1. Figure 7: According to the text and plectin levels depicted in Figure 6, it seems that the graphs were switch. Part A are probably results for 4T1 and part B for C6. There is no blocking experiment described, which would potentially confirm level of specific binding (vs. non-specific one). Please put kidney retention in the two graphs. There is no sense to put the kidney radioactivity retention in supplementary material.

Author reply: We thank the reviewer for his/her careful observation. We have corrected the results and put kidney retention in these two graphs. We also performed the in vivo blocking experiments, and got an interesting finding. Co-injection of PTP and 99mTc-HYNIC-PTP within a certain range seemed to decrease nonspecific accumulation in lung and kidney and enhance T/M ratio. This issue is still being investigated, and we plan to report the results in our further work.

Round 2

Reviewer 2 Report

The authors replied to the comments. I would suggest to include in the discussion the structural differences of the two molecules and how this difference can impact in their body distribution

Author Response

Reviewer 2:

The authors replied to the comments. I would suggest to include in the discussion the structural differences of the two molecules and how this difference can impact in their body distribution.

Author reply: We appreciate the reviewer for her/his good suggestion. According to the suggestion, some description was added in the discussion section of revised manuscript (Page 16 Line 476), also as shown below.

“Notably, both [99mTc]Tc-HYNIC-PTP and FITC-PTP showed distinct tumor uptake, while different biodistribution in liver and kidneys due to their structural differences. The hydrophilicity of HYNIC leaded to high accumulation of [99mTc]Tc-HYNIC-PTP in kidneys, but FITC-PTP had obvious fluorescence signal in liver, which was in line with the literature [30]. This suggested that the introduction of FITC or HYNIC probably changed the biodistribution and pharmacokinetics of PTP, with no significant influence on targeting ability.”
